# The Switching of Trailing Limb Anticipatory Locomotor Adjustments is Uninfluenced by what the Leading Limb Does, but General Time Constraints Remain

**Félix Fiset [1,2] and Bradford J. McFadyen [1,2,3,*]**

1    Centre for Interdisciplinary Research in Rehabilitation and Social Integration, CIUSSS-CN, IRDPQ, Québec City, QC G1M 2S8, Canada; felix.fiset.1@ulaval.ca
2    Department of Rehabilitation, Faculty of Medicine, Université Laval, Québec City, QC G1V 0A6, Canada
3    CIRRIS-CIUSSS-CN, 525 boulevard Hamel, Québec City, QC G1M2S8, Canada
*    Correspondence: brad.mcfadyen@fmed.ulaval.ca; Tel.: 418-529-9141



**Featured Application: The fundamental knowledge from this work has potential to eventually inform clinical interventions for gait and mobility training.**

**Abstract:** Research shows a blend of bilateral influence and independence between leading and trailing limbs during obstacle avoidance. Recent research also shows time constraints in switching leading limb strategies. The present study aimed to understand the ability to switch anticipatory locomotor adjustments (ALAs) in the trailing limb. Ten healthy young adults (24 ± 3 years) were immersed in a virtual environment requiring them to plan and step over an obstacle that, for the trailing limb, could change to a platform, requiring a switch in locomotor strategies to become a leading limb to step onto a new surface. Such perturbations were provoked at either late planning or early execution of the initial trailing limb obstacle avoidance. Sagittal plane trailing limb kinematics, joint kinetics and energetics were measured along with electromyographic activity of key lower limb muscles. Repeated measures ANOVAs compared dependent variables across conditions. To adjust to the new environment, knee flexor power around toe-off decreased ($p < 0.001$) and hip flexor power increased ($p < 0.001$) for late planning phase perturbations, while there was only an increase in mid-swing hip flexor power ($p < 0.05$) during perturbations at execution. Findings showed no influence of the leading limb function on the ability to switch trailing limb ALAs during late planning. However, the trailing limb was also constrained for modifying ALAs once execution began, but on-going limb control strategies could be exploited in a reactive mode.

**Keywords:** obstacle avoidance; anticipatory control; gait; locomotion; perturbation

---

## 1. Introduction

Navigating complex daily environments involves anticipatory control to safely avoid obstacles. There are robust and different lower limb strategies for stepping over an obstacle compared to stepping onto a platform. The anticipatory locomotor adjustment (ALA) to step over an obstacle involves a knee flexor generation strategy around toe-off (K5) [1,2] for both the leading and trailing limbs (respectively the first and second limbs to adapt) that puts energy in the lower limb to increase both knee and hip flexion. For the trailing limb, hip flexor generation power around toe-off (referred to as H3) for limb advancement can also be delayed until mid-swing (referred to as H3D) [3,4]. The higher the obstacle,

the greater H3D [3]. For a level change accommodation, the leading limb instead increases the existing hip flexor generation at toe-off to help lift the limb to the new height [5].

In addition, there are differences in the modes of control between the leading and trailing limbs. Mohagheghi et al. [6] showed that the leading limb control is modifiable online by visual information during execution, whereas the trailing limb depends on visuomotor memory. The authors also concluded that there is independent control between the leading and the trailing limbs related to this difference in the use of visual information. The visual memory of the trailing limb for obstacle avoidance has been shown to be robust enough to be retained during cessation of the task up to two minutes after the leading limb has crossed in humans [7].

The hypothesis of independence between the two lower limbs was also reinforced by unexpected perturbations due to contact with the obstacle and the fact that only the limb that made contact was modified on subsequent obstacle crossings [8]. More recently, Howe et al. [9] found that lead limb toe trajectory was affected by the combined loss of the lower half of the visual field and ankle skin local anesthesia, whereas the trailing limb was only affected by loss of lower visual input. This emphasizes differences in motor control between limbs. Finally, such differences in adapted limb control have been highlighted during memory-guided obstacle crossing wherein the obstacle was removed and only its position replaced by a contrast tape [10]. The success rate of the trailing limb was four times lower than for the leading limb. The authors suggested that greater dependence on visual sampling during the approach phase for the trailing limb could explain this degraded performance.

However, Santos et al. [11] suggested that the hypothesis of independence between leading and trailing limbs could have been observed because of the predictability of the static environments used. By changing obstacle dimension one step before clearance, the authors showed that the trailing limb can use visual information from the leading limb's obstacle crossing. In studying foot–body balance geometries for adapted gait, Dugas et al. [12] perturbed obstacle placement both early and late in the approach to and beginning of obstacle avoidance. Secondary findings of differences between clearance and maximum foot heights for the leading and trailing limbs across perturbation conditions further supported limb independence in leading and trailing limb adaptations within non-predictable environments.

While limb independence seems well accepted, the above-noted studies using dynamic environments also point to the ability to modify obstacle avoidance once it has been planned. In an early example of this, Patla et al. [13] showed that relatively quick modifications of a planned obstacle avoidance can be made when a second obstacle is introduced unexpectedly. Yet, all of this work only supports the modification of an already planned strategy. Very recently, McFadyen et al. [14] used virtual reality (VR) to instantly change environmental demands, requiring one to switch between the respective knee and hip strategies described earlier either during the late planning stage at foot contact preceding the obstacle or at the beginning of step execution for the leading limb. Results showed that leading limb strategies could be switched within the late planning stage but appeared to be "locked-in" once execution began, at least to allow the planned strategy to be initiated uninterrupted, regardless of whether substituting a knee with a hip strategy or vice versa.

However, given the presumed independence between leading and trailing limb control for obstacle avoidance noted above and differences in the underlying motor control, it is not clear whether the same time-constrained control previously shown [14] exists for the trailing limb. Therefore, the purpose of the present work was to study the ability to switch from a trailing limb obstacle avoidance strategy to a level change accommodation strategy with the leading limb during either late planning or early execution when following obstacle avoidance by the contralateral leading limb. Knowing underlying locomotor control constraints could be relevant to clinicians for mobility training in pathologic populations.

## 2. Materials and Methods

### 2.1. Participants

Ten healthy young adults (24 ± 3 years; 5 males; height = 1.75 ± 0.79 m; mass = 67.5 ± 9.0 kg) with normal or corrected-to-normal vision (Snellen chart eye test) and without musculoskeletal or neurological problems affecting their walking (self-reported) were recruited and provided their written consent to participate to the experiment. The study was approved by the ethics committee of the Centre intégré universitaire de santé et de services sociaux de la Capitale-Nationale (#2018-433).

### 2.2. Experimental setup and protocol

Non-collinear triads of markers were placed on the lateral aspects of the feet, lower legs and thighs and at the level of the sacrum for the pelvis. Markers were tracked by nine cameras (Vicon Motion Systems, Inc/MX T-series, Denver, USA; 100 Hz). Joint centers, segment centers of mass and radii of gyration were estimated in relation to anatomical landmarks (heels, toes, 5th metatarsal heads, medial/lateral malleoli, medial/lateral femoral condyles, left/right iliac spines and left/right anterior superior iliac spines) digitized with respect to the segment marker triads during a calibration set-up. It is unknown how limb dominance influences anticipatory locomotor adjustments (ALAs); and to control for any possible variability, the dominant limb, as determined from the Waterloo Footedness Questionnaire [15] (testing dominance during both movement and stabilization), was designated as the trailing limb for initial obstacle crossing.

Surface electromyography (EMG) was recorded (Noraxon, Scottsdale, USA; 1000 Hz) with electrodes on the mid-bellies of the rectus femoris (RF), vastus lateralis (VL), semitendinosus (ST), biceps femoris (BF), tibialis anterior (TA), medial gastrocnemius (GM) and soleus (SOL) muscles of the trailing limb. Signals were amplified (gain = 1000) and band-pass filtered at collection (15–500 Hz) and then offline (20–400 Hz). Signal strength and crosstalk were verified by isometric testing of each muscle, except for the GM and SOL, for which participants were asked to rise onto their toes while standing in order to solicit these muscles.

Participants donned a security harness used to prevent a complete fall. A starting line was determined at a point to allow the participant to make 3 steps at a self-selected natural speed with the trailing foot landing on a force plate (Bertec Corporation, Columbus, USA, 1000 Hz) corresponding to the final position before clearing an obstacle (1.6 cm$^2$ wooden stick; Figure 1A). A second environment, involving the same starting position, introduced a 1.5 m deep platform (Figure 1B) positioned at 0.71 m (obstacle front to platform front) from where the obstacle was placed. Both surface changes were set to heights between 15% to 16% of participants' lower limb lengths. Participants were familiarized with both physical environments performing five trials of stepping over an obstacle and five trials to step onto the platform. Five additional trials of stepping over an obstacle with the leading limb only and then stepping onto the platform with both limbs (Figure 1C) were carried out to practice the experimental conditions requiring them to adjust for an obstacle with the leading limb first and a level change to the platform with the trailing limb immediately after. For all trials, participants were asked to walk at a natural pace.

After these trials in the real world, participants were fitted with a head mounted display (HMD, Oculus Rift V1, Menlo Park, USA) tracked by three non-colinear markers. A virtual reality system similar to the one used previously [14] produced a virtual environment (VE) resembling the laboratory with the above described surface changes (Figure 1A,B). This VE was modeled in Blender and programmed with Vizard (WorldViz, Inc, Santa Barbara, USA) to be synchronized to the participant's movements. The lower limbs of a first-person avatar were also aligned with those of the participant's, using a skeletal template from the Vicon data to allow participants to also see their lower limbs in real time (refresh rate of 60 Hz). This was important for more realistic control of lower limbs during ALAs.

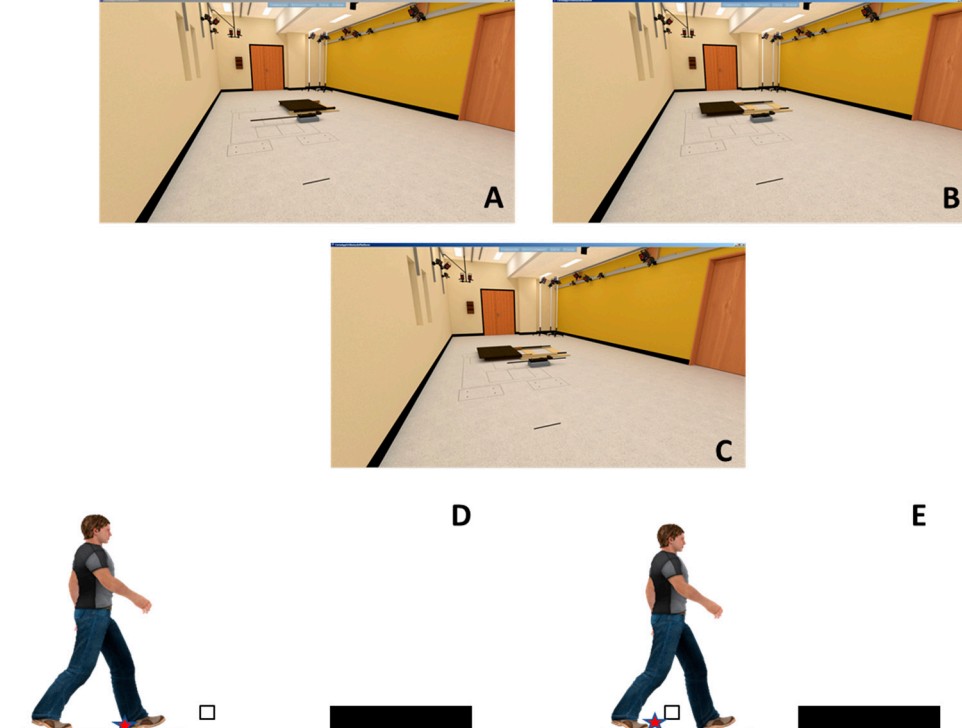

**Figure 1.** Virtual environment with the obstacle (**A**), platform (**B**) and the half obstacle and platform (**C**). The timing of instantaneous perturbations was triggered by a 20 N load at trailing limb heel contact (**D**; T1) or at 20 N of unloading at trailing limb toe-off (**E**; T2).

After participants were comfortably fitted in the HMD, both real obstacle and platform environments were also set in place and simultaneously presented in the VE, and participants were invited to step over the obstacle, step onto the platform and touch them both with a foot for as long as they needed to feel comfortable and to understand that the VE coincided spatially to the real environment. Participants then finally practiced both obstructed environments individually as the ALA tasks to be used in the protocol.

During data collection, participants were required to approach and step over the obstacle in the VE that could disappear and be instantaneously replaced with a platform appearing 71 cm further along the path requiring a new locomotor strategy by the trailing limb. Two experimental conditions, presented in blocks counterbalanced across participants, were related to the timing of this VE change during the trailing limb step preceding the obstacle at either foot contact (Figure 1D) or at the subsequent toe-off (Figure 1E). The environment changes were respectively triggered when the vertical ground reaction force under the trailing foot exceeded 20 N (T1 corresponding to late planning) and falling below 20 N (T2 corresponding to execution). Each block was comprised of a total of 15 trials, including 5 environment perturbations and 10 catch trials without environment perturbations. Participants were instructed to walk at a comfortable and natural rhythm and to adapt their gaits to the new environment if presented. Participants were assured that the physical environment always corresponded to the last environment that was presented in the HMD. To provide confidence to the participants, before collection of each experimental block, participants first observed the instantaneous change from obstacle to platform and then performed three practice trials for the environment, knowing it would change at the respective time. Between all data collection trials, the image in the HMD was blacked out and participants wore earphones that played pink noise at a volume determined to block all ambient noise within the laboratory to prevent them from anticipating the upcoming condition.

At the end of data collection, French versions of the simulator sickness questionnaire [16] were used to measure simulator sickness, and the Presence Questionnaire [17] and SUS questionnaire [18], both used to measure the degree of VE immersion, were administered.

*2.3. Data analysis*

Marker, force plate and full wave rectified EMG data were all passed through lowpass, zero lag, Butterworth filters with cut-off frequencies set respectively to 6, 50 and 100 Hz based on previous work [14,19]. Relative joint angles were calculated using Cardan rotation matrices prioritizing the sagittal plane. Maximum relative joint angles (MJRA) during the ALA swing phase, minimum foot clearance (MFC; the vertical distance between the heel and the obstacle or platform front edge) and mean COM velocity (COMv; as the mean velocity in sagittal plane of pelvis COM during the ALA stride) were also calculated. Newton–Euler inverse dynamics equations with a custom-made program were used to estimate lower limb kinetics and energetics respectively for net muscle moments of force (MMF) and muscle mechanical power (product of the MMF and relative joint angular velocities). These data were normalized to body mass, and muscle mechanical work of the trailing limb was calculated as the mathematical integral of targeted positive and negative power bursts corresponding to ankle plantar flexor generation (push-off; A2), knee extensor absorption at the end of stance (K3), the delayed knee extensor absorption around mid-swing (K3D), knee flexor generation at toe-off (K5) and hip flexor generation at both toe-off (H3) and mid-swing (H3D). Hip hiking work (HH) around toe-off was also calculated as the area under the positive power burst from the product of the vertical hip joint reaction force and vertical hip joint velocity with positive power corresponding to energy from the pelvis.

The areas under muscle bursts for BF and ST EMG activity that aligned with the K5 power burst for obstacle avoidance [1] were calculated. As in a previous study [14], we took electromechanical delays into consideration by choosing these muscle bursts to begin at the local minimum prior to the K5 power burst and continue for the duration found for the corresponding K5 power burst of that trial. This allowed EMG data to provide physiological confirmation of knee flexor strategies across conditions. To compare across participants, percentage changes in these EMG bursts from the mean unperturbed trials for each perturbation time within each environment block were also calculated. As there was no appropriate hip flexor muscle activity recorded (RF has not been shown to be a good measure of a hip flexor activity [20]), no corresponding EMG areas were presented for hip flexion.

For descriptive time series data, EMG, kinematic, kinetic and energetic (muscle mechanical power) data were normalized to 100% of stride, with toe-off fixed at 60% and ensemble averaged across trials for each condition.

*2.4. Statistical analysis*

Repeated measures ANOVAs for perturbation conditions were used (SPSS, IBM, New York, USA, v.24) to compare the three perturbation timings (unperturbed, early and late). Sphericity was verified for each variable with Mauchly's test of sphericity, and a Huynh–Feldt correction was applied when appropriate. When main effects were found, Bonferroni-corrected post-hoc comparisons were applied for further comparisons between conditions. A Wilcoxen signed-rank test was used to compare percentage changes in targeted knee flexor bursts between T1 and T2 perturbated conditions. Significance level was set to p = 0.05 (including after Bonferroni corrections) for all analyses. Effect sizes (partial eta squared, $\eta^2$) with 95 % confidence intervals (CI; {lower endpoint, upper endpoint}) are presented with ANOVA data.

**3. Results**

It appears that all participants tolerated the VE well given average low group scores of 0.31/3 ± 0.19 on the Simulator Sickness Questionnaire. In addition, average group scores of 5.51/7 ± 0.77 were found on the Presence Questionnaire, and they were 4.54/7 ± 0.68 on the SUS Questionnaire, showing that participants were relatively well immersed.

Two participants contacted the platform without tripping when it appeared at trail foot toe-off (T2). One of them contacted it twice and the other only once. These trials were kept for analysis because the participants did not lose their balance and patterns prior to platform contact did not show any outlying behavior.

Before presenting the dependent variables, the general pattern changes within the time series patterns for angular displacement, net muscle moment of force and net muscle power plots are described. Across the three joints, locomotor adjustments for the new VE were mostly seen around toe-off and during the swing phase (Figure 2). At the ankle, a greater dorsiflexion occurred at mid-swing when the obstacle changed to the platform at trailing foot toe-off (T2). The related kinetic and energetic changes were less evident in the plots because of the relatively small mass of a foot to control during swing phase compared to the whole body during the stance phase.

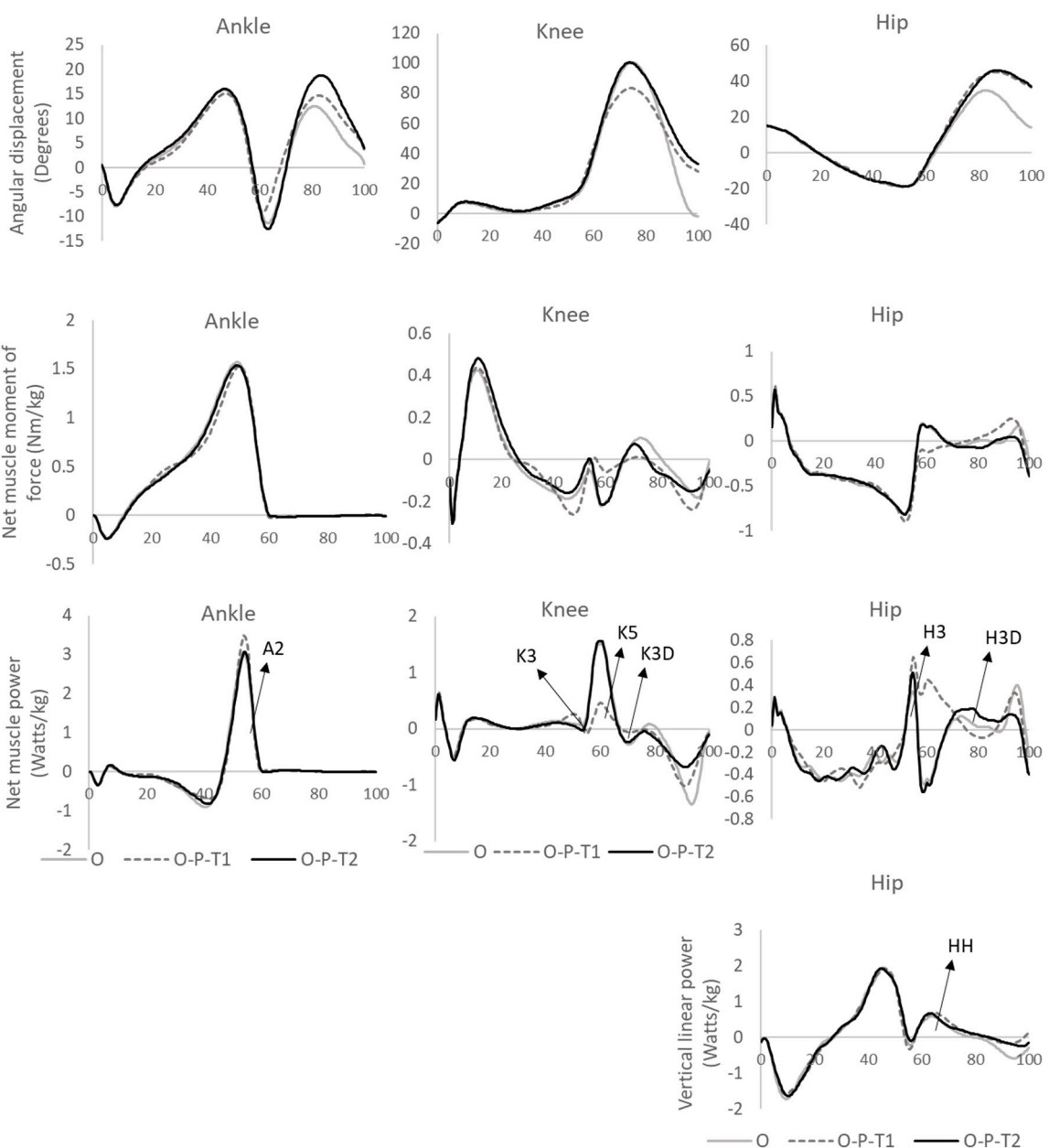

**Figure 2.** Average plots of angular displacement (top), net muscle moment of force (middle) and net muscle power (bottom) at the ankle (left), knee (middle) and hip (right) joints of the trailing limb during unperturbed conditions (gray line), and perturbations of obstacle to platform at T1 (O-P-T1; dashed line) and T2 (O-P-T2; black line). Gait stride was normalized to 100% with toe-off fixed at 60%.

At the knee joint, there was a decrease in knee flexion combined with a decreased knee flexor moment associated with decreased generation for the early change (T1) at toe-off. For the late perturbation (T2), kinetic and energetic curves were similar to those for the obstacle condition.

Finally, at the hip joint, hip flexion was higher when the obstacle changed to a platform, regardless of the timing of perturbations (T1 or T2). The kinetic and energetic plots showed a larger hip flexor moment combined with energy generation around toe-off (H3) when the platform appeared early (T1). For the obstacle-only condition and late perturbation (T2), there was some hip flexor energy generation at toe-off that switched to a hip extensor moment with energy absorption before returning, at mid-swing, to hip flexor power generation (H3D). There did not appear to be any change in linear hip hiking power (HH) across conditions.

To better understand how an obstacle avoidance strategy can change to level accommodation strategy, specific kinematic variables were calculated and presented in Figure 3. First, MJRA of dorsiflexion, knee and hip flexion during the obstacle crossing swing phase were all significantly different across conditions with a main effect of time ($p < 0.001$ for the three joints; ankle $\eta^2 = 0.6811$ CI{0.3308, 0.7906}; knee $\eta^2 = 0.8117$ CI{0.5647, 0.8761}; hip $\eta^2 = 0.8340$ CI{0.6107, 0.8907}). More precisely, at the ankle, dorsiflexion was greater for late perturbation than O ($p < 0.001$) and O-P-T1 ($p = 0.007$) conditions, while O-P-T1 was similar to obstacle condition ($p = 0.715$). At the knee, MJRA were only decreased for the early perturbation compared to the obstacle condition ($p < 0.001$) and to late perturbation ($p < 0.001$), but the late perturbation was not different from the obstacle condition ($p = 1.000$). At the hip, MJRA increased for both changes to Pl ($p < 0.001$ for T1 and T2), and no differences were observed between the two perturbation timings ($p = 1.000$).

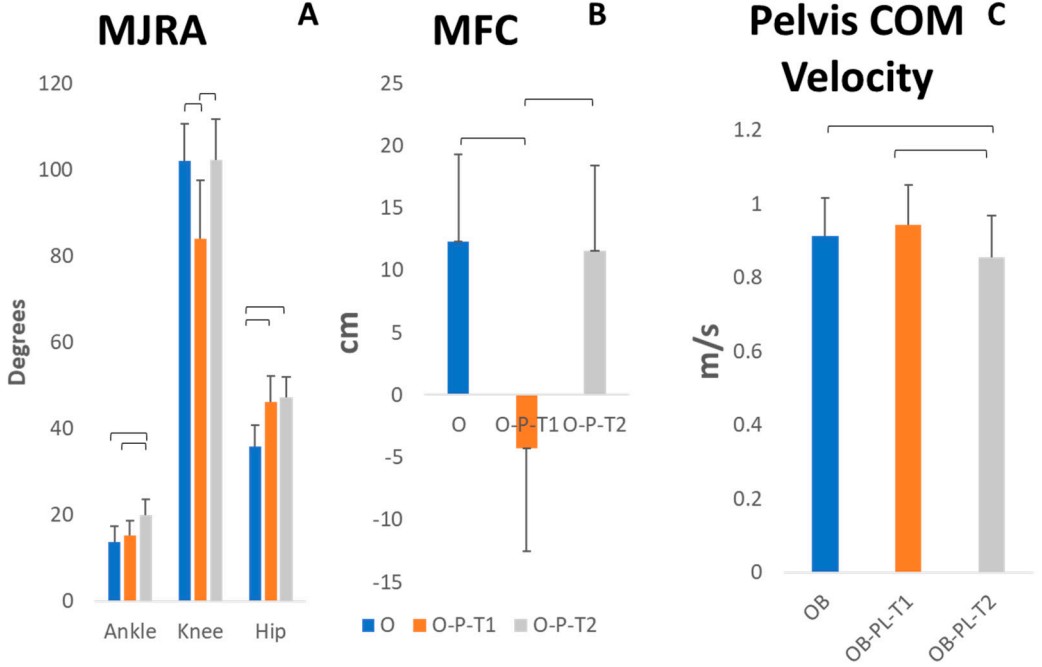

**Figure 3.** Average (standard deviation) of maximum joint relative angles (MJRA) during trailing limb swing phase (**A**). Minimal trailing limb foot clearance (MFC) of the obstacle—negative values correspond to the foot passing below obstacle level (**B**). Average pelvis center of mass (COM) velocity during trailing limb swing phase (**C**) across unperturbed (blue), early perturbation of obstacle to platform (O-P-T1, orange) and late perturbation (O-P-T2, gray) conditions.

Finally, MFC of the trailing foot over the obstacle had a main effect of time ($p < 0.001$; $\eta^2 = 0.7874$ CI{0.3939, 0.8743}) with decreased MFC for early VE change compared to obstacle only and to the T2 perturbation ($p < 0.001$ for both), and late perturbation was not different than obstacle only ($p = 1.000$). A main effect of time was also present for COMv ($p = 0.002$; $\eta^2 = 0.4987$ CI{0.1064, 0.6677}). COMv was

significantly slower for O-P-T2 condition than for the obstacle condition ($p = 0.008$) and for O-P-T1 ($p = 0.025$), but these latter two conditions were not different from each other ($p = 0.673$). We can see in the plot of COMv (Figure 4) that participants appeared to slow down only during the swing phase.

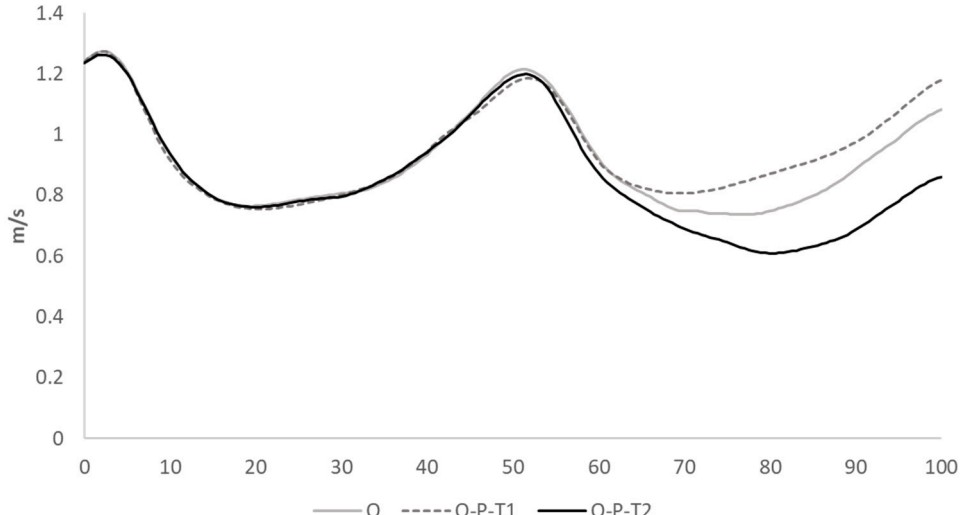

**Figure 4.** Average plots of pelvis COM velocity (COMv) across unperturbed (gray), early perturbation of obstacle to platform (O-P-T1, dashed line) and late perturbation (O-P-T2, black line) conditions. Gait stride was normalized to 100% with toe-off fixed at 60%.

For the accommodation strategies related to the muscle mechanical work for the power bursts around toe-off and during swing (Figure 5), ankle push-off (A2) remained unchanged across conditions ($p = 0.755$; $\eta^2 = 0.028$ CI{0, 0.1939}). Knee extensor absorption at the end of the stance phase (K3), however, showed a main effect of time ($p = 0.014$; $\eta^2 = 0.4357$ CI{0.0216, 0.6544}). Post-hoc analyses showed a tendency of K3 to be augmented for early change ($p = 0.053$), but not for late one ($p = 0.146$), and the O-P-T1 (obstacle-platform at T1) condition was not different than O-P-T2 conditions ($p = 0.141$). For knee flexor generation (K5) and the subsequent delayed knee extensor absorption (K3D), there were main effects of time for both power bursts ($p < 0.001$; $\eta^2 = 0.7894$ CI{0.3962, 0.8756}, $p = 0.001$; $\eta^2 = 0.6356$ CI{0.1944, 0.7786} respectively). Post-hoc analyses showed that K5 significantly decreased when the obstacle switched to a platform early (O-P-T1; $p < 0.001$), but not late (O-P-T2; $p = 1.000$). Significant differences were also observed between the two perturbation timings ($p < 0.001$). The delayed knee extensor absorption right after K5 (K3D) was also significantly decreased for the early change compared to the obstacle only ($p = 0.002$) and late perturbation conditions ($p = 0.019$). There was no difference between late perturbated trials and obstacle only ($p = 1.000$). Hip flexor generation around toe-off (H3) showed a general effect of time ($p < 0.001$; $\eta^2 = 0.7530$ CI{0.3192, 0.8555}) and was augmented in an early perturbation (O-P-T1; $p < 0.001$), but not for a late one (O-P-T2; $p = 0.408$). O-P-T1 and O-P-T2 conditions were also different from each other ($p = 0.003$). There was also a main effect of time for the hip flexor generation during mid-swing (H3D; $p = 0.005$; $\eta^2 = 0.4685$ CI{0.0637, 0.6578}). Post-hoc analyses showed that it was increased for the late perturbation only (O-P-T2) compared to the obstacle condition ($p = 0.037$) and to the O-P-T1 condition ($p = 0.034$) with no significant difference observed between O-P-T1 and obstacle conditions ($p = 0.379$ ).The positive hip linear power burst at toe-off showed a main effect of time ($p = 0.009$; $\eta^2 = 0.4830$ CI{0.0925, 0.6568}) with an increase of power burst for a late perturbation compared to the obstacle condition ($p < 0.001$). The obstacle condition was not different from O-P-T1 ($p = 0.519$). O-P-T1 was not different from O-P-T2 ($p = 0.242$).

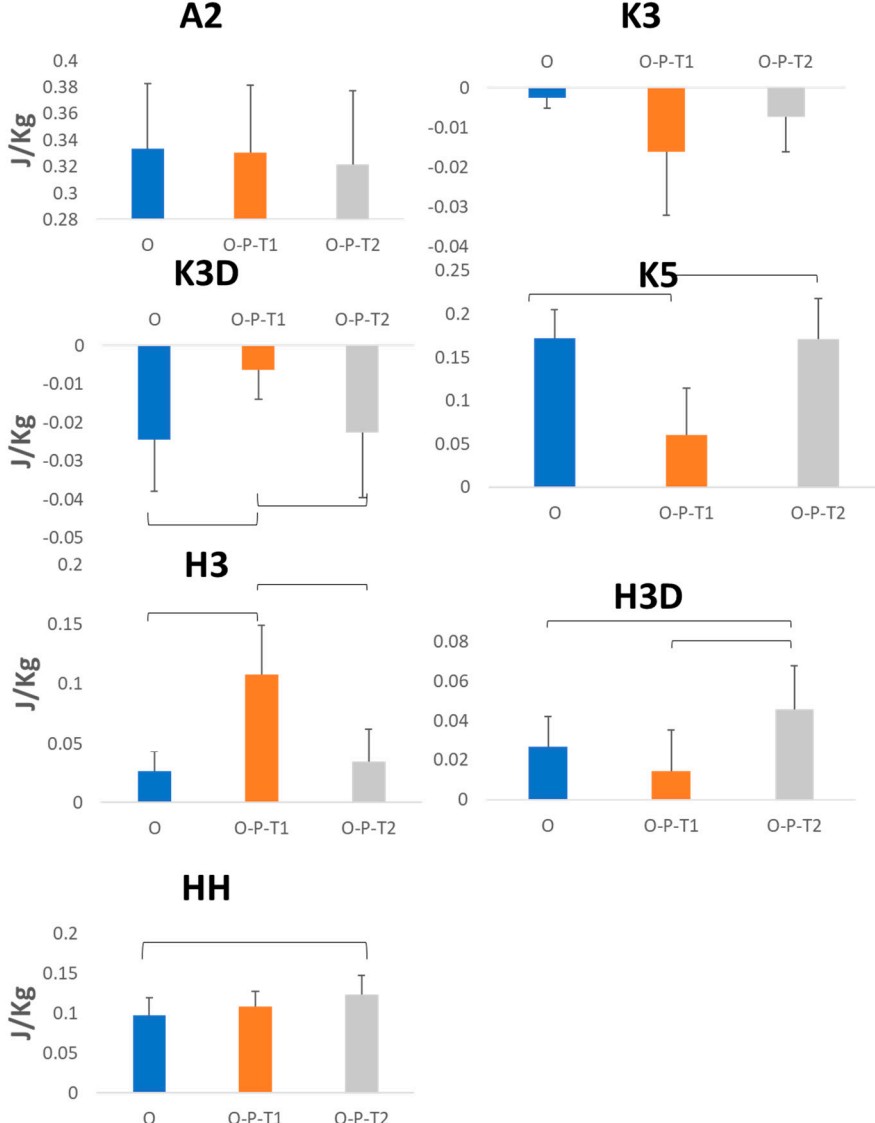

**Figure 5.** Average (standard deviation) of net muscle work corresponding to ankle push-off (A2); knee extensor absorption around (K3); after toe-off (K3D); after knee flexor generation (K5); hip flexor generation around toe-off (H3); hip flexor generation at mid-swing (H3D); and hip hiking around at toe-off (HH) across the unperturbed (O; blue), early perturbation of obstacle to platform (O-P-T1, orange) and late perturbation (O-P-T2, gray) trials.

The related muscle activity is presented in Figure 6 for a participant with patterns representing what was observed across all participants, specifically for the areas of interest for statistical testing. EMG plots of tibialis anterior showed increased activity in swing during late perturbation (T2). Semitendinosus (ST) and the biceps femoris (BF) activity showed higher activity at toe-off, corresponding to the K5 muscle power burst, for the obstacle and late perturbation (T2) conditions. All other muscles appeared to be qualitatively similar across conditions. To confirm the qualitative observations for greater ST and BF activity at toe-off corresponding to the K5 power burst, a percentage of change relative to obstacle condition was calculated for each perturbation condition (Figure 7). For both targeted bursts of ST and BF, muscle activity for the early perturbation condition was lower than for the late one ($p = 0.017$ for ST; $p = 0.007$ for BF).

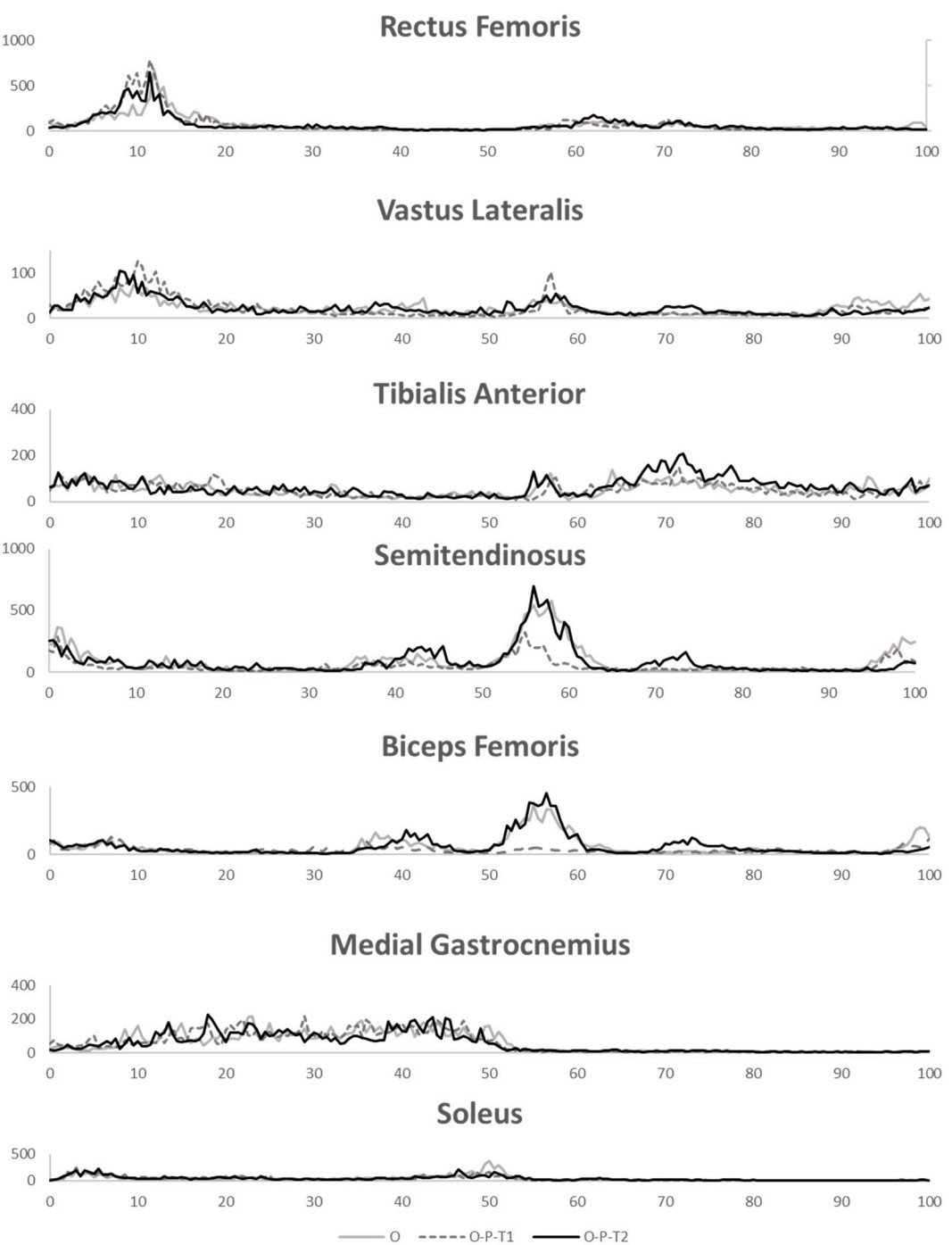

**Figure 6.** Surface EMG of seven lower limb muscles of one representative subject across conditions of unperturbed (O; gray), early perturbation of obstacle to platform (O-P-T1, dashed line) and late perturbation (O-P-T2, black line) conditions.

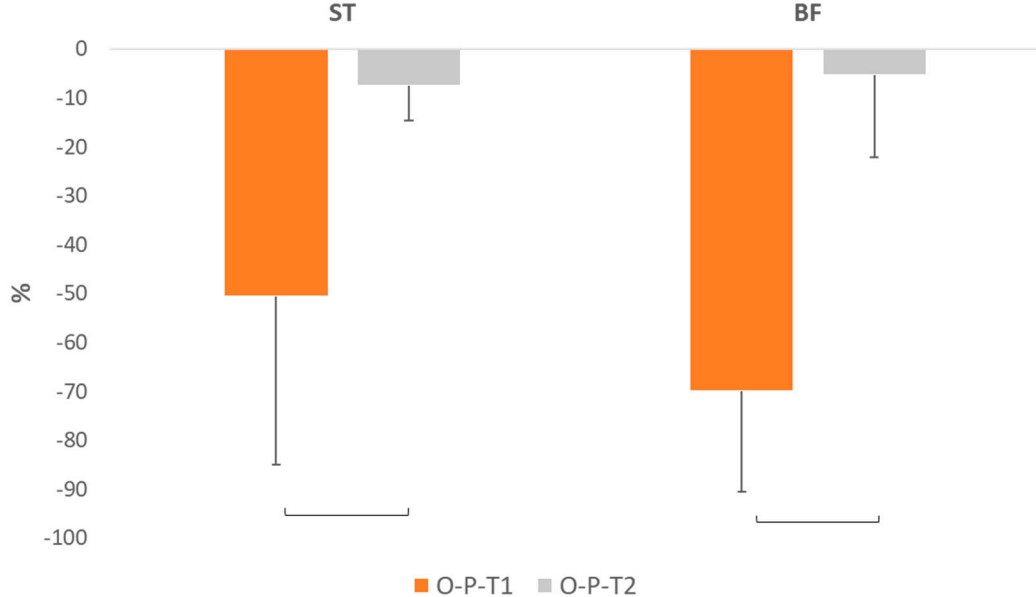

**Figure 7.** Percentage changes in the area under the curve of semitendinosus (ST) and biceps femoris (BF) EMG activity corresponding to the K5 power burst across the early perturbation of obstacle to platform (O-P-T1, orange) and late perturbation (O-P-T2, gray) conditions.

## 4. Discussion

ALAs are essential to maintaining equilibrium within the dynamic environments found daily. Earlier work has shown the capacity to add onto an already planned ALA online [13], while a more recent study has revealed that one can also switch from one ALA to another for the leading limb, but only before ALA execution begins [14]. Yet, there is ample research to suggest differences in sensorimotor control between leading and trailing limbs [6,7]. The present study involved a perturbation requiring participants to switch from an initially planned obstacle avoidance ALA in the trailing limb to a leading limb level change accommodation ALA. This strategy switch created a contrast between what the leading and trailing limb planned and executed. As previously observed by McFadyen et al. [14], it was also found here that a trailing limb ALA can be switched within the late planning stage, but not once execution begins. In addition, it was found that, differently than for the leading limb, the pre-existing hip flexor work at the transition to swing phase could be exploited after initial execution to accommodate the need to step up to a higher surface level.

The current observation of the ability to switch trailing limb ALAs during late planning (T1) at the same time as the leading limb executes obstacle avoidance, supports the fact that the leading limb obstacle avoidance function was of insignificant influence on the trailing limb successfully switching to a new ALA. Such an ALA switch must, therefore, be driven by visual information of the environmental change. This ability to use visual information for trailing limb control has already been suggested for a static obstacle avoidance situation when vision was manipulated during planning and approach [6]. In addition, Santos et al. [11] showed this in a more dynamic situation for changing obstacle heights at a trailing limb foot contact (similar timing to T1 in the present study). However, previous studies involved only modifying an already planned trailing limb strategy. In the present work, ALAs after switching were very different for both limbs, and the leading limb knee flexor strategy to step over the obstacle did not affect the capacity to switch ALAs within the trailing limb. This further supports independent control between limbs [6,8,12]. It appears that the previously suggested feedforward influence of leading limb function on trailing limb control [11] is thus context specific and likely only true when both limbs perform the same ALA (i.e., both stepping over an obstacle).

Despite this lack of influence of leading limb control at the early switch (T1) of ALAs in the trailing limb, it was observed that the initially planned obstacle avoidance strategy could not be canceled out

at the beginning of its execution (T2). As noted above, this is probably not due to influence from the leading limb and is more likely for the same reason discussed by McFadyen et al. [14] for switching leading limb ALAs. Specifically, a locomotor strategy is not immediately modifiable once it is launched regardless of being a leading or trailing limb. This inability to switch as execution begins (T2) was clearly expressed in the lower limb kinematic, kinetic and EMG patterns with large effect sizes as shown by the partial eta squared values for most variables.

Yet, despite the fact that ALAs could not switch at the beginning of the execution, some modifications in locomotor patterns were seen at mid-swing. More precisely, for the T2 perturbation, the trailing limb increased the already existing mid-swing delayed hip flexor generation (H3D) seen for trailing limb obstacle avoidance. This was likely to compensate for the inability to increase the pre-planned hip flexor generation at toe-off (H3) for obstacle avoidance. This may be similar to what Patla et al. [13] proposed as an ability to exploit an on-going plan (in their case for knee flexion of the leading limb) and modulate it online at a later latency. That is, the delayed hip flexor generation at mid-swing would was already planned for trailing limb obstacle avoidance, but then further exploited to replace lost H3 strategy for a platform ALA. In a previous study [14], when the leading limb switched at execution, a hip extensor burst appeared during swing phase 300 ms after the environment perturbation. This timing is similar to the fine-tuning stage of gait adaptations [6] and was considered to be more a reactive response rather than a strategy substitution. The present study shows a similar reaction time for the trailing limb for the increased H3D in mid-swing. This reactive strategy would also be assisted by a slight increase in hip hiking power (HH) as well representing a frontal plane contribution. Yet, this latter change on its own was at the same time as when the initially planned obstacle ALA was first carried out. Thus, it would appear that HH is autonomous to the lower limb ALA and can be added on at early latency, even when the newly erroneous obstacle ALA must be initiated for the late perturbation at execution. The observed significant changes in H3D and HH power bursts are supported by good effect sizes with approximately half the variance explained by the conditions. However, with the larger confidence intervals, some caution is required.

Results at the ankle joint also showed greater mid-swing dorsiflexion and TA activity for the late perturbation condition (T2) but without changes immediately following toe-off. This again supports no immediate means to switch an ALA once executed, but an ability to adapt online at a later latency as a reactive strategy. Why the ankle creates more dorsiflexion than the other conditions even if it appears to finish the stride on the new surface level with the same relative joint angles between T1 and T2 perturbations is unclear. It could be a preventative overshoot to avoid tripping in this reaction to the new environment.

It is difficult from the present study to deduce the underlying neural control. With respect to the same observations for lead limb ALA switching, McFadyen et al. [14] offered some conjecture from the literature. They noted that perhaps motor plans involving posterior parietal cortex (PPC) [21] could lead to muscle synergies set in the motor cortex through pyramidal tract neurons [22]. These synergies may be initially locked-in upon execution, but can also be modified at a later point, perhaps through reticular spinal neuron pathways suggested to be involved in pattern modifications [23]. In the present study, wherein participants had to switch from a trailing limb obstacle crossing strategy to a leading limb level accommodation strategy, it is highly likely that similar neural control is involved. However, while PPC would likely be involved in trailing limb motor planning, it appears to also be important for the visual memory of the obstacle for this limb [24,25]. However, this visual memory would be useless in the current ALA switch. PPC cell discharge is related to bilateral function rather than any specific limb. Thus, it could perhaps reorganize motor planning such that cells related to trailing limb function would be inhibited while other cells related to the new leading limb function would need to newly discharge in relation to the new position of the limb relative to the platform that appeared. The premotor cortex (PMC) might also be involved in this transition from trailing to leading limb roles given recent literature suggesting some limb-dependent cells of PMC in cats that contribute the selection of the first limb to step over and the related interlimb coordination [26]. How such neural

pathways are be related to the constraints in anticipatory and reactive control of trailing and leading limb function for ALAs discussed above will require more study.

Although more research is required to understand any clinical relevance of these present findings, given the differences in the underlying control of the leading and trailing limbs, it could be clinically relevant to train each side in leading and trailing functions to improve dynamic balance. As to the inability to switch ALAs once execution begins, but with online adaptation of hip flexors to assure final and safe accommodation, perhaps reactive training could be another point of focus.

The present study is not without limitations. Virtual reality is a great way to control environmental factors and produce protocols not possible in the physical environment. However, it is important participants see it as being as real as possible. The results showed acceptable presence levels, but as discussed in previous similar work [14], VR did also result in some behavior (higher foot clearance and maintaining some knee flexor generation power (K5) for a platform accommodation) that is different from what might be expected in an only physical environment. However, in this within-subject design, we are confident that we were able to show general control changes between the ALA switching conditions. The present study was also limited in number of participants and to healthy young adults only. Further work is needed to understand such adaptability constraints and their consequences for safe ALAs in other populations (e.g., older adults, populations with neurological impairments).

## 5. Conclusions

The present study involved switching from a planned trailing limb obstacle avoidance involving a knee flexor strategy to a leading limb surface level accommodation involving a hip flexor strategy. The findings showed that, despite presumed feedforward control involving the leading limb during the original plan, such leading limb influence appears context specific, and the trailing limb can benefit from ongoing visual information. However, there also appears to be a limb independent rule that anticipatory locomotor adjustments can be switched during the planning phase, but at execution, the planned pattern must be initiated. Yet, the control system appears able to exploit other aspects of the ongoing lower limb control strategies, such as the hip flexor power at mid-swing, to accommodate to the new locomotor demands of the environment. Underlying neural mechanisms for such flexibility and constraints in anticipatory locomotor adjustments are only speculative for now and require further study. More research is also needed to better understand the implications of such control for locomotor control changes with age and impairments.

**Author Contributions:** Conceptualization, methodology, data curation: B.J.M. and F.F.; formal analysis, writing—original draft preparation: F.F.; writing—review and editing, supervision, project administration, funding acquisition: B.J.M. All authors have read and agreed to the published version of the manuscript.

**Funding:** Research was funded Natural Sciences and Engineering Research Council of Canada (RGPIN/191782-2017).

**Acknowledgments:** The authors thank N. Robitaille (programming), S. Forest (infrastructure construction), G. St-Vincent and F. Dumont (motion capture) for valuable technical assistance and Dr. Jean Leblond for statistical consulting. We also thank Dr. Philippe Jackson for the use of the HMD.

**Conflicts of Interest:** The authors declare no conflict of interest.

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
