# Peer review of "The Switching of Trailing Limb Anticipatory Locomotor Adjustments is Uninfluenced by what the Leading Limb Does, but General Time Constraints Remain"

_applsci, doi:10.3390/app10072256_

Round 1

Reviewer 1 Report

The introduction is perhaps too extensive. Some of the information in the introduction could be properly placed in the discussion. Please summarize the introductory ideas concisely.

The work is interesting and well designed. The statements in the discussion may be more cautious, due to the small sample size.

It would be interesting if in the future they conducted a randomized clinical trial with a control group and a higher sample size, to reach more reliable conclusions.

Author Response

We thank the reviewers for seeing the value of the work and for their rapid and detailed reviews to help us improve the manuscript. Please find below our response to each comment. Revisions in the manuscript for all reviewers have been indicated by tracked changes.

REVIEWER 1:

The introduction is perhaps too extensive. Some of the information in the introduction could be properly placed in the discussion. Please summarize the introductory ideas concisely.

REPONSE: We find ourselves with a compromise to add information requested by the other reviewers and cut the introduction as suggested here. However, we did try to be more concise in the revised version and have also removed the part about animal work to move it to the discussion only.

The work is interesting and well designed. The statements in the discussion may be more cautious, due to the small sample size.

REPONSE: Although the sample size is not large, it is similar to many studies of this type. We have noted this limit however, and, in response to other reviewers, have added more statistical support for the findings.

It would be interesting if in the future they conducted a randomized clinical trial with a control group and a higher sample size, to reach more reliable conclusions.

REPONSE: The present work is still very fundamental with the goal to understand mechanisms, but, yes, it would be interesting to take this further, even as an intervention after evidence has been proven.

Reviewer 2 Report

This manuscript describes a study that uses VR to create changes in presence and absence of obstacles and raised platforms at different times in the gait cycle. Subjects' muscle strategies and limb movements are measured.

Comments for consideration are listed below:

Define the acronym ALA in the abstract

Page 2, Line 2: Change tailing to trailing

Page 2, Line 25: Change non-predicable to non-predictable

Page 3, Line 10: Bipedal gaits are symmetric, so there aren’t true leading and trailing limbs like there are in quadrupedal asymmetric gaits likely cantering or galloping. I’d strongly suggest the authors consider defining leading and trailing limbs early on in the manuscript, specifically with respect to tasks like stepping over an obstacle or stepping onto a platform. The limb dominance seems to be important when considering these results, but is briefly mentioned once in the methods. “The dominant limb was determined from the Waterloo Footedness Questionnaire [20] and designated as the trailing limb for obstacle crossing.” This could be highlighted more for readers. Is the dominance with respect to mobilizing or stabilizing in the Waterloo Questionnaire?

Page 6: Most results are stated without any statistical evidence. Please state that these are qualitative observations, or provide magnitude of differences with p-values.

Many results refer to a main effect of time. What other effects were considered in the ANOVA? Please state all effects included in the ANOVA model in the methods section.

Figure 5 legend: Revise “across the of unperturbed”

Figures switch between using different colors to label 3 groups, black/gray/dashed line and blue/gray/orange. Use consistent colors for each group across all figures.

Page 12, Line 21, 25, 28, 29: Change “trail” to “trailing”. Check the entire text for similar instances of “lead” and “leading”. Be consistent.

Page 14, Line 6: Change “what might be expected for real” to “what might be expected outside of VR”

Page 14, Line 7: Explain “single-subject design”. There were 10 subjects in this study. Are you referring to repeated measures? How was subject considered in the ANOVA model? Also, within the methods, please address whether the residuals of the ANOVA were normally distributed, which is an assumption of the test.

Author Response

We thank the reviewers for seeing the value of the work and for their rapid and detailed reviews to help us improve the manuscript. Please find below our response to each comment. Revisions in the manuscript for all reviewers have been indicated by tracked changes.

REVIEWER 2:

Define the acronym ALA in the abstract

RESPONSE: This is now defined at the beginning of the abstract.

Page 2, Line 2: Change tailing to trailing

RESPONSE: This has been done

Page 2, Line 25: Change non-predicable to non-predictable

RESPONSE: This has been done

Page 3, Line 10: Bipedal gaits are symmetric, so there aren’t true leading and trailing limbs like there are in quadrupedal asymmetric gaits likely cantering or galloping. I’d strongly suggest the authors consider defining leading and trailing limbs early on in the manuscript, specifically with respect to tasks like stepping over an obstacle or stepping onto a platform. The limb dominance seems to be important when considering these results, but is briefly mentioned once in the methods. “The dominant limb was determined from the Waterloo Footedness Questionnaire [20] and designated as the trailing limb for obstacle crossing.” This could be highlighted more for readers. Is the dominance with respect to mobilizing or stabilizing in the Waterloo Questionnaire?

RESPONSE: We added definitions in parenthesis at the first mentioned of leading/trailing limbs in the introduction.

To our knowledge, it is not known if limb dominance affects the execution of stepping over an obstacle or onto a platform. We simply focused on the dominant limb for our study to avoid any potential effects and this been clarified in the methods.

Half of the questions in the Waterloo footedness questionnaire assessed the dominance with respect to mobilizing and the other half for stabilizing. This is perfect for us given because during an obstacle avoidance task, the trailing limb is in a stabilization role during leading limb execution and then in a mobilizing role during obstacle crossing. The added detail for the questionnaire has been added.

Page 6: Most results are stated without any statistical evidence. Please state that these are qualitative observations or provide magnitude of differences with p-values.

RESPONSE: Yes, the time-series results are descriptive only and used to set up the patterns and understand the dependent variables. This is now mentioned on page 6. We also identified the power bursts (A2, K3, K3D, K5, H3, H3D, HH) on the figures to allow the reader to easily match the quantitative analyses to these plots.

Many results refer to a main effect of time. What other effects were considered in the ANOVA? Please state all effects included in the ANOVA model in the methods section.

RESPONSE: We used a repeated measure ANOVA with one factor. This has been clarified in the methods.

Figure 5 legend: Revise “across the of unperturbed”.

REPONSE: We have now clarified this to: “across the unperturbed”.

Figures switch between using different colors to label 3 groups, black/gray/dashed line and blue/gray/orange. Use consistent colors for each group across all figures.

RESPONSE: Time series graphs are now all in black and grey. Colors were kept for histograms.

Page 12, Line 21, 25, 28, 29: Change “trail” to “trailing”. Check the entire text for similar instances of “lead” and “leading”. Be consistent.

RESPONSE: Modifications have been made where appropriate to be consistent.

Page 14, Line 6: Change “what might be expected for real” to “what might be expected outside of VR”

RESPONSE: This has been done

Page 14, Line 7: Explain “single-subject design”.

RESPONSE: Our apologies. Actually, it is repeated measures ANOVA, as discussed in the methodology and we meant a “within subject design”. The correction has been made.

There were 10 subjects in this study. Are you referring to repeated measures? How was subject considered in the ANOVA model? Also, within the methods, please address whether the residuals of the ANOVA were normally distributed, which is an assumption of the test.

RESPONSE: It is difficult to verify normality with the number of participants involved. However, we made distribution (quartile-quartile) plots for each dependent variable across conditions and these showed that the distribution of averages was close to a normal distribution except for the O-P-T1 condition for foot clearance due to the fact that one participant had a higher clearance than the others. Yet this was not an outlying behaviour and the participant changed foot clearance the same way across conditions as the other participants and was not different for any other variable. Furthermore, homogeneity of variances was already tested using the Mauchly test for sphericity and a Huynh-Feldt correction was made when necessary. This information has been added to the manuscript in the description of statistical analyses. Finally, in response to Reviewer 3, partial Eta squared values (for effect sizes) and intervals of confidence have been added to the results as well.

Reviewer 3 Report

General Comments:

The purpose of the manuscript was to study the ability to switch from a trailing limb obstacle voidance anticipatory locomotor adjustment to a leading limb strategy  for level change accommodation during either late planning or early execution when following obstacle avoidance by the leading limb. The authors presented a good justification for this study, identifying a gap in the existing literature. Some interesting results are presented, although there are a number of potential methodological issues which should be addressed / improved. Further, the writing / presentation style should be improved in places. The paper has potential but more information is required around some of my concerns before deciding whether it is possible to correct them.

Specific Comments:

Abstract:

Line 16: How many participants? Also remove the decimal places for age as it is expressed as an integer.

Lines 23-24: ALA has not yet been defined here. Please try to avoid abbreviations in sections such as the abstract, aims, and conclusions as many readers may go straight to these sections and they should be easy to read alone.

Some results would improve the abstract – e.g. some statistical results.

Introduction:

Throughout the manuscript many unnecessary abbreviations are used. Abbreviations such as Ob and Pl replace single words and make the manuscript very difficult to read in places. Please try to limit the use of abbreviations as much as possible throughout the text to only those that are absolutely necessary.

Please justify your font rather than left alignment throughout

P1 Line 34: The term ‘hip flexor generation’ is unclear

P1 Line 35: Please be more specific – rather than saying it depends on obstacle height, does it increase or decrease with increases in height?

P2 L4: Are studies in horses and cats relevant if the information has already been presented in humans? What is Area 5 and what are the implications of this?

P2 L12: ‘Has been’ should be ‘have been’?

P2 L13-16: You previously said that the trail limb was more memory based than the lead limb – does this research contradict that?

P2 L32: Can the hip and knee strategies be described in more detail as these are important throughout the paper? Are these two alternatives or two extremes of a continuum?

P2 L34: How are the late planning stage and early execution defined?

P2 L40-41: It is unclear that you are referring to the same limb when saying trailing limb and leading limb.

Overall, this is a good introduction. Please add a little about the applications or implications of such a study – i.e. why the gap in the literature should be filled. Were you testing a specific hypothesis based on the previous literature?

Materials and Methods:

P2 L45: Is 10 participants enough? What is the power of the study for the analyses performed? Please remove decimal places for years and report height in m.

P3 L5: Where exactly were these markers positioned? A figure may help.

P3 L7: What do you mean by ‘digitized’ landmarks?

Please discuss the limitations of sEMG (pros and cons) with specific reference to your application. For example, please see: Vigotsky, A. D., Halperin, I., Lehman, G. J., Trajano, G. S., & Vieira, T. M. (2018). Interpreting signal amplitudes in surface electromyography studies in sport and rehabilitation sciences. Frontiers in physiology, 8, 985.

P3 L17: Can a better explanation be given for ‘rise on their toes’?

P3 L24: ‘the’ isn’t needed in ‘familiarized with the both’ – please proof read the whole manuscript to check for similar errors as there are a few.

P4 L12: Do you have a reference to support this last sentence?

P4 L20: Is the 71 cm measured from front-to-front of the obstacle / platform?

The terms T1, T2, etc could be made more intuitive, which would make the manuscript much easier to read. For example, an abbreviation that links to what it actually describes

P5 L5: Is there likely to be any effect of the participants knowing a change is likely compared to being truly unaware?

P5 L11: Please justify each of the cut-off frequencies. Why 6 Hz for kinematics – residual analysis, etc? Why has the GRF been filtered, and if so why not at the same frequency as the kinematics? E.g. Kristianslund, E., Krosshaug, T., & van den Bogert, A. J. (2012). Effect of low pass filtering on joint moments from inverse dynamics: implications for injury prevention. Journal of biomechanics, 45(4), 666-671.

What are the effects of isolating the sagittal plane kinematics? Are adjustments likely to occur outside of this plane?

Can more intuitive names be given to terms such as K3, K3D, HH, H3D, etc? This makes some sections very difficult to read later on.

P5 L36: What is meant by energetics?

P5 L39: Did the ANOVAs use ensemble averaged data or data from individual trials? Were maximums from individual trials or averages (e.g. peak of means or mean of peaks)? How many data points were included in each analysis and is this sufficient?

P5 L43: Please report effect sizes and confidence intervals and interpret these to inform your discussion and conclusion rather than relying on p values alone.

Results:

P6 L1-3: Are there any references or standards for what is classed as ‘low’ or ‘good’?

P6 L3: Please add a space after the plus/minus.

P6 L9 – P8 L2: There are a lot of descriptive / qualitative comparisons within here. The figures provide a lot of good information but this is not interpreted statistically. It is therefore unclear which differences are significant and/or meaningful. For example, you could compare the entire time series using statistical parametric mapping (code available at spm1d.org) – this would enable you to say which normalised time regions were significantly different or not. If interested in a hip or knee dominant strategy then you could use vector coding to assess the proximal-distal coordination pattern – e.g.: Floría, P., Sánchez-Sixto, A., Harrison, A. J., & Ferber, R. (2019). The effect of running speed on joint coupling coordination and its variability in recreational runners. Human movement science, 66, 449-458. These two techniques can even be combined to see whether the proximal –distal coordination significantly differs between conditions at particular normalised time points.

P8 L7: Please add spaces either side of < or =

P8 L9: Please report each comparison or make it clear if other comparisons are not significant and so not discussed. In some cases you report 2 out of the 3 post-hoc comparisons only.

Has any correction been performed for the multiple tests? i.e. have the p values for individual ANOVAs been corrected for the fact that multiple ANOVAs are performed?

Figure 4: This is an excellent example of where statistical parametric mapping may show no significant differences from 0 to around 70% but then differences from around 70 – 100% of time.

Figure 6: Please check the colours in all figures looks okay in grayscale.

Figure 7: The title can be removed as there is already a figure title below. Are there any significant differences in this figure?

Discussion:

P12 L12: This doesn’t make sense or read very well – please could it be rewritten?

P12 L19+: This paragraph is very clear and well-written.

P13 L9+: This is an example of a paragraph with far too many abbreviations that is therefore difficult to read and interpret.

P13 L19: Should the d be removed from the end of ‘staged’?

P13 L42: ‘known shown’ – should one word be removed?

P14 L5 – 7: ‘Real’ can be reworded as ‘real as possible’ doesn’t read well/

P14 L7: Single-subject design – you have 10 subjects. Do you mean ‘within-subject design’?

Conclusions:

It would be good to include a brief explanation of how the adjustments are made (using the kinetic and kinematic results for example).

Author Response

We thank the reviewers for seeing the value of the work and for their rapid and detailed reviews to help us improve the manuscript. Please find below our response to each comment. Revisions in the manuscript for all reviewers have been indicated by tracked changes.

REVIEWER 3:

General Comments:

The purpose of the manuscript was to study the ability to switch from a trailing limb obstacle voidance anticipatory locomotor adjustment to a leading limb strategy for level change accommodation during either late planning or early execution when following obstacle avoidance by the leading limb. The authors presented a good justification for this study, identifying a gap in the existing literature. Some interesting results are presented, although there are a number of potential methodological issues which should be addressed / improved. Further, the writing / presentation style should be improved in places. The paper has potential but more information is required around some of my concerns before deciding whether it is possible to correct them.

Response: we are pleased you found the work of interest. Please see below for our detailed response to your comments.

Specific Comments:

Abstract:

Line 16: How many participants? Also remove the decimal places for age as it is expressed as an integer.

RESPONSE: We now indicate the number of participants et removed decimals for the average age.

Lines 23-24: ALA has not yet been defined here. Please try to avoid abbreviations in sections such as the abstract, aims, and conclusions as many readers may go straight to these sections and they should be easy to read alone.

RESPONSE: ALA is now defined in the abstract. It is written as a full word in the declaration of study objectives and maintains as an acronym in the rest of the manuscript.

Some results would improve the abstract – e.g. some statistical results.

RESPONSE: This sentence is added to the abstract: “To adjust to the new environment, knee flexor power around toe-off decreased (p<0.001) and hip flexor power increased (p<0.001) for late planning phase perturbations while there was only an increase in mid-swing hip flexor power (p<0.05)during execution.

Introduction:

Throughout the manuscript many unnecessary abbreviations are used. Abbreviations such as Ob and Pl replace single words and make the manuscript very difficult to read in places. Please try to limit the use of abbreviations as much as possible throughout the text to only those that are absolutely necessary.

RESPONSE: We have replaced the abbreviations for Ob and Pl with full words (note that we did not highlight these as there were too many. We kept them however for the figures (O, O-P-T1 and O-P-T2) to identify the conditions.

Please justify your font rather than left alignment throughout

RESPONSE: Done.

P1 Line 34: The term ‘hip flexor generation’ is unclear

RESPONSE: We meant the hip flexor generation power related to the energetic data, more precisely the H3 power burst discussed in the cited literature and later in the manuscript.

P1 Line 35: Please be more specific – rather than saying it depends on obstacle height, does it increase or decrease with increases in height?

RESPONSE: The delay remains the same, but the proportion of delayed hip flexor power generation increased with obstacle height. We have added the text: “The higher the obstacle, the greater H3D [3].

P2 L4: Are studies in horses and cats relevant if the information has already been presented in humans? What is Area 5 and what are the implications of this?

RESPONSE: Yes, it is relevant to talk about this animal work because it helps to understand neurologic structures involved in working memory in obstacle avoidance. This is the goal of the paragraph near the end of the discussion. However, we removed this reference to animal work in the introduction now.

P2 L12: ‘Has been’ should be ‘have been’?

RESPONSE: Done

P2 L13-16: You previously said that the trail limb was more memory based than the lead limb – does this research contradict that?

REPONSE: No, we do not think that our findings contradict the use of visual memory for trailing limb control. It shows that it is more important for the trail limb to view the obstacle during the approach phase. In fact, it appears to add the fact that memory can be put a side for a new strategy. Present perturbation trials were previously practiced so participants were likely able to use visual information to switch from one memorized environment to another when the perturbation occurred during late planning. However, during late perturbation at the beginning of execution, working memory of the previously unchanged environment was already put in motion and therefore not immediately modifiable.

P2 L32: Can the hip and knee strategies be described in more detail as these are important throughout the paper? Are these two alternatives or two extremes of a continuum?

RESPONSE: There are two set strategies described in the literature, one to step over an obstacle, and another one to step onto a platform (See McFadyen et Carnahan, 1997). We added information in the first paragraph of the introduction for the knee flexor strategy. We feel that hip strategy for stepping up was clear. We also mention the identifying acronyms from the literature (K5 for knee strategy and H3 with hip strategy) at this point.

These strategies are more of a continuum. For example, from our results, we saw that when the participants increased the hip strategy, knee strategy decreased, or vice-versa.

P2 L34: How are the late planning stage and early execution defined?

RESPONSE: We have attempted to be clearer in both the introduction and in the methods that this is at the step preceding the obstacle. The pre-existing sentences that follow this text indicate the forces used to define the perturbation timing.

P2 L40-41: It is unclear that you are referring to the same limb when saying trailing limb and leading limb.

RESPONSE: This sentence has been reformulated. We were talking about the same limb.

Overall, this is a good introduction. Please add a little about the applications or implications of such a study – i.e. why the gap in the literature should be filled. Were you testing a specific hypothesis based on the previous literature?

RESPONSE: We have added the reference to the first sentence of the last paragraph of the introduction related to previous literature and added a sentence at the end of this paragraph in reference to some potential clinical applications of this work.

Materials and Methods:

P2 L45: Is 10 participants enough? What is the power of the study for the analyses performed? Please remove decimal places for years and report height in m.

REPONSE: A pre-study power analysis was not performed, but we maintained the same number of participants as in similar work just published (McFadyen et al, 2018) and seen for many like-studies. We understand this is not necessarily a rigorous response. Yet, post-hoc power analyses are not to be encouraged, but given our p values and the fact that “observed power” values were high for significant findings (lowest for K3 power at 0.7310 and ranging between 0.8648 to 0.9999 for all other variables), we feel confident that the data are able to support the discussion. Of course, we do also already note the limitation of the number of participants in the manuscript.

P3 L5: Where exactly were these markers positioned? A figure may help.

RESPONSE: We hesitate to add another figure to the work, but we did add additional details concerning marker positions in the text.

P3 L7: What do you mean by ‘digitized’ landmarks?

RESPONSE: We meant the numerical collection of the additional markers placed on some anatomical landmarks for initial calibration. We reformulated this sentence to make it clearer.

Please discuss the limitations of sEMG (pros and cons) with specific reference to your application. For example, please see: Vigotsky, A. D., Halperin, I., Lehman, G. J., Trajano, G. S., & Vieira, T. M. (2018). Interpreting signal amplitudes in surface electromyography studies in sport and rehabilitation sciences. Frontiers in physiology, 8, 985.

RESPONSE: EMG data were used only to support the strategies found by the inverse dynamics data. Given this complementary nature and secondary focus on this sEMG data, we did not discuss such limitations, but if the reviewer insists, we can add a section to the limitations text.

P3 L17: Can a better explanation be given for ‘rise on their toes’?

RESPONSE: We have modified the sentence. It now reads: “… to rise onto their toes while standing in order to solicit these muscles

P3 L24: ‘the’ isn’t needed in ‘familiarized with the both’ – please proof read the whole manuscript to check for similar errors as there are a few.

RESPONSE: We apologize for these errors and we have carefully proof read the manuscript.

P4 L12: Do you have a reference to support this last sentence?

RESPONSE: It was not clear which lines of the manuscript are being referred too, but if you mean “The lower limbs of a first-person avatar were also aligned with those of the participant’s, …. This was important for more realistic control of lower limbs during ALAs”, we don’t have any related literature for this type of protocol, but we tested this in the laboratory ourselves and found that stepping over and up without vision of the lower limbs was more difficult.

P4 L20: Is the 71 cm measured from front-to-front of the obstacle / platform?

RESPONSE: Yes, it was measured from front-to-front. We added this information.

The terms T1, T2, etc could be made more intuitive, which would make the manuscript much easier to read. For example, an abbreviation that links to what it actually describes

RESPONSE: We prefer to keep the terms T1 and T2 as they were used in another paper with similar perturbations for the leading limb (McFadyen et al., 2018), but we moved them to the sentence referring to the triggering to make the link to the “T” and have also related them to planning and execution.

P5 L5: Is there likely to be any effect of the participants knowing a change is likely compared to being truly unaware?

RESPONSE: It is possible that knowing that a change could occur, could have an effect on behaviour, yet participants were still unable to change strategies at T2, suggesting there was no anticipation and reinforcing our conclusion that the trailing limb is constrained in modifying ALAs once execution begins.

P5 L11: Please justify each of the cut-off frequencies. Why 6 Hz for kinematics – residual analysis, etc? Why has the GRF been filtered, and if so why not at the same frequency as the kinematics? E.g. Kristianslund, E., Krosshaug, T., & van den Bogert, A. J. (2012). Effect of low pass filtering on joint moments from inverse dynamics: implications for injury prevention. Journal of biomechanics, 45(4), 666-671.

REPONSE: We certainly understand this point. However, in this within-subject design during gait with low impact forces (the work you refer found greatest errors for high impact, fast movements), we felt that the cut-off frequency previously justified by Winter (1990, Biomechanics and Motor Control of Human Movement, Wiley and Sons) did not affect the overall findings of the work. Further proof of this is the concordance between knee power data and muscle burst changes from sEMG.

What are the effects of isolating the sagittal plane kinematics? Are adjustments likely to occur outside of this plane?

RESPONSE: The targeted strategies at the hip and knee as described in the literature are within the sagittal plan [1-5]. Some adjustments probably also occur in other planes, and the hip hiking presented in this work is related to frontal plane work. We now note that hip hiking represents frontal plane action.

Can more intuitive names be given to terms such as K3, K3D, HH, H3D, etc? This makes some sections very difficult to read later on.

RESPONSE: While we appreciate the reviewer’s point, particularly for those not familiar with these abbreviations, they do have a long history in such gait publications (including our own) and we have defined them. However, we have now also added them to the time-series plots to provide additional visual definitions.

P5 L36: What is meant by energetics?

REPONSE: Energetics here refers to the muscle mechanical power and related work (area under the power curves). We have noted this relationship along with kinetics for muscle moments of force in the methods.

P5 L39: Did the ANOVAs use ensemble averaged data or data from individual trials? Were maximums from individual trials or averages (e.g. peak of means or mean of peaks)? How many data points were included in each analysis and is this sufficient?

RESPONSE: Only the time-series curves were normalized in time and ensembled averaged for descriptive analysis. All quantitative analyses were done with averages across five trials for each individual from non-normalized data. Thus, with 10 participants averaged across 3 conditions, there were 30 data points for each variable. We feel that with the fact that general behaviour was the same for all participants and the results were clear for our key variables (e.g., K5 and H3) that this was enough.

P5 L43: Please report effect sizes and confidence intervals and interpret these to inform your discussion and conclusion rather than relying on p values alone.

RESPONSE: A sentence noting the use of Partial Eta Squared (η2) with 95 % confidence intervals has been added in the methodology and these values have been added across the results section.

Results:

P6 L1-3: Are there any references or standards for what is classed as ‘low’ or ‘good’?

RESPONSE: To our knowledge, there are no references. However, we changed the text to downplay this for the presence scores. We believe we can still say 0.31/3 is low for the SSQ.

P6 L3: Please add a space after the plus/minus.

RESPONSE: Done

P6 L9 – P8 L2: There are a lot of descriptive / qualitative comparisons within here. The figures provide a lot of good information but this is not interpreted statistically. It is therefore unclear which differences are significant and/or meaningful. For example, you could compare the entire time series using statistical parametric mapping (code available at spm1d.org) – this would enable you to say which normalised time regions were significantly different or not. If interested in a hip or knee dominant strategy then you could use vector coding to assess the proximal-distal coordination pattern – e.g.: Floría, P., Sánchez-Sixto, A., Harrison, A. J., & Ferber, R. (2019). The effect of running speed on joint coupling coordination and its variability in recreational runners. Human movement science, 66, 449-458. These two techniques can even be combined to see whether the proximal –distal coordination significantly differs between conditions at particular normalised time points.

RESPONSE: We thank the reviewer for the suggestion. However, the goal of this descriptive presentation of the time-series data was to orient the reader to the patterns presented as strategies from the quantified power bursts. We don’t feel the suggested analyses (although interesting) are necessary. The power bursts have also now been identified on the time-series figures to better orient the reader to the following statistical analyses.

P8 L7: Please add spaces either side of < or =

RESPONSE: Done

P8 L9: Please report each comparison or make it clear if other comparisons are not significant and so not discussed. In some cases you report 2 out of the 3 post-hoc comparisons only.

RESPONSE: We added the comparisons that were not mentioned. We are now reporting the 3 post-hoc comparisons.

Has any correction been performed for the multiple tests? i.e. have the p values for individual ANOVAs been corrected for the fact that multiple ANOVAs are performed?

RESPONSE: Yes, Bonferroni corrections were applied in post-hoc analyses. This was mentioned, and we also now note that significance levels were set to p = 0.05 after Bonferroni corrections.

Figure 4: This is an excellent example of where statistical parametric mapping may show no significant differences from 0 to around 70% but then differences from around 70 – 100% of time.

Figure 6: Please check the colours in all figures looks okay in grayscale.

RESPONSE: Done.

Figure 7: The title can be removed as there is already a figure title below. Are there any significant differences in this figure?

RESPONSE: Good point. We also forgot to put the significant differences on the figure. This is now added.

Discussion:

P12 L12: This doesn’t make sense or read very well – please could it be rewritten?

RESPONSE: It has been rewritten.

P12 L19+: This paragraph is very clear and well-written.

RESPONSE: Thank you.

P13 L9+: This is an example of a paragraph with far too many abbreviations that is therefore difficult to read and interpret.

RESPONSE: We think it is now better with the abbreviations for Ob and Pl removed as noted above. We also defined H3, H3D and HH again to help the reader.

P13 L19: Should the d be removed from the end of ‘staged’?

RESPONSE: Yes, we remove it.

P13 L42: ‘known shown’ – should one word be removed?

RESPONSE: We removed shown.

P14 L5 – 7: ‘Real’ can be reworded as ‘real as possible’ doesn’t read well/

RESPONSE: Done

P14 L7: Single-subject design – you have 10 subjects. Do you mean ‘within-subject design’?

RESPONSE: Yes, we meant within-subject design. The correction has been done.

Conclusions:

It would be good to include a brief explanation of how the adjustments are made (using the kinetic and kinematic results for example).

REPONSE: We added some more information about the adjustments made.

Round 2

Reviewer 3 Report

General Comments:

The authors have made a clear attempt to modify the manuscript in response to each of my previous comments and I believe the manuscript has improved as a result. They should be commended for this. I still have some minor concerns but I believe these can be addressed. Other than some issue around statistical reporting and interpretation, the specific comments are mostly very minor.

Specific Comments:

Abstract:

Line 16: Space before years

Lines 23-25: I’d prefer a space either side of < but this may be personal preference

Keywords: All are separated by ; but the last two separated by a comma

Introduction:

Line 34: Is there a space before the full stop? It seems odd on a new line.

Line 78: ‘Locked-in’ seems quite an informal term – can this be better described?

Line 85: ‘When following obstacle avoidance by the contralateral limb’ may be better than ‘leading limb’ to avoid confusion between limbs during this sentence.

Methods:

Line 90: Space in 0.79m

Line 96: Should ‘legs’ be ‘shanks’ to specify the lower leg?

Lines 115-116: I don’t believe the ‘Ob’ abbreviation for ‘obstacle’ or ‘Pl’ for ‘platform’ are necessary here anymore. All abbreviations used in tables and figures should be explained in the table or figure heading anyway so that they can be read on their own.

Fig 1: Fig 1D and 1E don’t add anything – it’s clear what contact and toe-off mean, and these figures don’t add more information as they’re simply the same figure twice with a red star in different positions.

Lines 165-166: I accept your justification for not filtering kinetics and kinematics at the same cut-off frequency, but please include within the text the manner in which these particular frequencies were selected.

Statistical Analysis: In what software were statistical analyses performed?

Line 202: Cohen’s d would be better than partial eta squared. I know SPSS usually gives partial eta squared but this can easily be converted.

Results:

Line 208: Something has gone wrong with the ‘_0 .68’. The same odd underscore occurs elsewhere (e.g. lines 244 – 250).

Qualitative results: I understand that the purpose of these results is purely descriptive. However, the graphs are used to make claims such as ‘there was an earlier dorsiflexion’,  ‘a greater dorsiflexion occurred’, ‘there was a decrease in knee flexion’, etc. The vast majority of these claims can be statistically tested. Even if you are unwilling to analyse the entire time curve then you could compare discrete parameters such as timings or magnitudes. This is not your primary statistical analysis and is not hypothesis-driven beforehand and so should never be presented as such. It would be a secondary exploratory analysis in light of your qualitative suggestions. Figure 4 is a great example of where the qualitative description is used to support or shine further light on the quantitative results, rather than the other way around.

Lines 244 – 245: A space either side of the equals would improve this

Line 292: This line is just one example but please be consistent around the use of 0.X or .X

Line 304: How do the EMG plots for this subject compare to others? A statement around the inter-individual variation would help here – i.e. help readers to know whether these same patterns could be expected in all or most subjects.

Discussion:

Lines 405-406: How was the behaviour different to what might be expected in a physical only environment? Can this be related back to your results?

You added to the Introduction that ‘Knowing underlying locomotor control constraints could be relevant to clinicians for mobility training in pathologic populations’. Can you now discuss these practical applications in light of your findings?

You have now added effect sizes and confidence intervals but these are not discussed in the Discussion. It is not clear how they have informed the manuscript. Are the effects large enough to be meaningful (e.g. clinically or practically meaningful)? What is the level of uncertainty around the parameter estimates?

Author Response

Please find below our response to Reviewer 3. Modifications to the manuscript are indicated by blue highlights. Please also note that all previous tracked changes from version 1 have been accepted unless related to Reviewer 3’s comments below. Finally, our responses are also provided within the context of the time limit provided by the journal (5 days starting March 18) and the constraints of distancing from the research centre and laboratory imposed by our institute in the current global situation of COVID-19. That is, we don’t have direct access to the data base at the moment. However, we feel we have been able to respond to all points and have a even stronger manuscript.

Specific Comments:

Please note that very minor changes (e.g., punctuation) and corrections were done as required, but not highlighted throughout the manuscript.

Abstract:

Line 16: Space before years

RESPONSE: done

Lines 23-25: I’d prefer a space either side of < but this may be personal preference

Keywords: All are separated by ; but the last two separated by a comma

RESPONSE: done

Introduction:

Line 34: Is there a space before the full stop? It seems odd on a new line.

RESPONSE: Fixed

Line 78: ‘Locked-in’ seems quite an informal term – can this be better described?

RESPONSE: This term certainly exists in a medical diagnosis of “locked-in syndrome”. The on-line Merriam-Webster dictionary cites it as an adjective. We have put quotations around it.

Line 85: ‘When following obstacle avoidance by the contralateral limb’ may be better than ‘leading limb’ to avoid confusion between limbs during this sentence.

RESPONSE: added “contralateral” but left “leading”

Methods:

Line 90: Space in 0.79m

RESPONSE: Fixed

Line 96: Should ‘legs’ be ‘shanks’ to specify the lower leg?

RESPONSE: We have changed it to lower leg.

Lines 115-116: I don’t believe the ‘Ob’ abbreviation for ‘obstacle’ or ‘Pl’ for ‘platform’ are necessary here anymore. All abbreviations used in tables and figures should be explained in the table or figure heading anyway so that they can be read on their own.

RESPONSE: These were overlooked and have been removed

Fig 1: Fig 1D and 1E don’t add anything – it’s clear what contact and toe-off mean, and these figures don’t add more information as they’re simply the same figure twice with a red star in different positions.

RESPONSE: Well we do not dispute the comment made by the reviewer, we feel a visual guide helps and complements the environmental depictions above it. As this does not affect content, we prefer to let the journal decide if they want this figure reduced.

Lines 165-166: I accept your justification for not filtering kinetics and kinematics at the same cut-off frequency, but please include within the text the manner in which these particular frequencies were selected.

RESPONSE: We have included references to previous work.

Statistical Analysis: In what software were statistical analyses performed?

RESPONSE: Information added

Line 202: Cohen’s d would be better than partial eta squared. I know SPSS usually gives partial eta squared but this can easily be converted.

RESPONSE: While we respect the opinion of the reviewer, this work and responses to review have been guided by a trusted statistician at our centre. Given this was not raised in the initial review by any reviewers and that partial eta squared provides the information sought we opted to keep this variable, but, as noted in our response to another comment below, we have added some discussion of the meaning of these values.

Results:

Line 208: Something has gone wrong with the ‘_0 .68’. The same odd underscore occurs elsewhere (e.g. lines 244 – 250).

RESPONSE: These were only the tracked changes to show the addition of a space that was requested in the first review. These marks are removed when accepted.

Qualitative results: I understand that the purpose of these results is purely descriptive. However, the graphs are used to make claims such as ‘there was an earlier dorsiflexion’,  ‘a greater dorsiflexion occurred’, ‘there was a decrease in knee flexion’, etc. The vast majority of these claims can be statistically tested. Even if you are unwilling to analyse the entire time curve then you could compare discrete parameters such as timings or magnitudes. This is not your primary statistical analysis and is not hypothesis-driven beforehand and so should never be presented as such. It would be a secondary exploratory analysis in light of your qualitative suggestions. Figure 4 is a great example of where the qualitative description is used to support or shine further light on the quantitative results, rather than the other way around.

RESPONSE: It is true we did not test for the earlier dorsiflexion timing, but we did present data for all of the other descriptions (see figures 3 and 5 and related text). Given that this dorsiflexion timing is not discussed and within the time limits of the journal and the limited access to the laboratory in the current global context, we have simply removed this part and only refer to data supported further by statistics. We don’t believe this affects the impact of the results.

Lines 244 – 245: A space either side of the equals would improve this

RESPONSE: Fixed

Line 292: This line is just one example but please be consistent around the use of 0.X or .X

RESPONSE: We changed all to 0.X.

Line 304: How do the EMG plots for this subject compare to others? A statement around the inter-individual variation would help here – i.e. help readers to know whether these same patterns could be expected in all or most subjects.

 RESPONSE: While there are individual differences in muscle activity, the “representative subject” was chosen because they represented the overall patterns observed for all subjects and, more specifically, represented the dependent variables shown by all participants, but modified with some inter-participant variability as shown in Fig 7. Again, Figure 6 is only to guide the reader to understand the patterns behind the statistical analyses made. We have attempted to clarify the text on page 10 lines 306-307.

Discussion:

Lines 405-406: How was the behaviour different to what might be expected in a physical only environment? Can this be related back to your results?

RESPONSE: We discussed differences between physical and virtual environments in our previous work [14]. We have added reference to this on lines 420-21.

You added to the Introduction that ‘Knowing underlying locomotor control constraints could be relevant to clinicians for mobility training in pathologic populations’. Can you now discuss these practical applications in light of your findings?

RESPONSE: A small paragraph has now been added to the end of the discussion preceding the limits section.

You have now added effect sizes and confidence intervals but these are not discussed in the Discussion. It is not clear how they have informed the manuscript. Are the effects large enough to be meaningful (e.g. clinically or practically meaningful)? What is the level of uncertainty around the parameter estimates?

RESPONS: Partial Eta squared show that most of the variance were explained by our conditions in our statistical findings. We believe that the effect size is large enough to be meaningful. However, the confidence intervals were larger for H3D and HH with a smaller lower limit. We now note this in the discussion.